# Zein-Based Nanoparticles as Oral Carriers for Insulin Delivery

**DOI:** 10.3390/pharmaceutics14010039

**Published:** 2021-12-24

**Authors:** Cristian Reboredo, Carlos J. González-Navarro, Ana Luisa Martínez-López, Cristina Martínez-Ohárriz, Bruno Sarmento, Juan M. Irache

**Affiliations:** 1Department of Chemistry and Pharmaceutical Technology, University of Navarra, C/Irunlarrea 1, 31008 Pamplona, Spain; creboredo@alumni.unav.es (C.R.); amlopez@unav.es (A.L.M.-L.); 2Center for Nutrition Research, School of Pharmacy and Nutrition, University of Navarra, C/Irunlarrea 1, 31008 Pamplona, Spain; cgnavarro@unav.es; 3Department of Chemistry, University of Navarra, C/Irunlarrea 1, 31008 Pamplona, Spain; moharriz@unav.es; 4i3S, Instituto de Investigação e Inovação em Saúde, Alfredo Allen 208, 4200-180 Porto, Portugal; bruno.sarmento@i3s.up.pt; 5CESPU—Instituto de Investigação e Formação Avançada em Ciências e Tecnologias da Saúde, 4585-116 Gandra, Portugal

**Keywords:** zein, insulin, nanoparticles, mucus-permeating, poly(ethylene glycol), oral delivery

## Abstract

Zein, the major storage protein from corn, has a GRAS (Generally Regarded as Safe) status and may be easily transformed into nanoparticles, offering significant payloads for protein materials without affecting their stability. In this work, the capability of bare zein nanoparticles (mucoadhesive) and nanoparticles coated with poly(ethylene glycol) (mucus-permeating) was evaluated as oral carriers of insulin (I-NP and I-NP-PEG, respectively). Both nanocarriers displayed sizes of around 270 nm, insulin payloads close to 80 µg/mg and did not induce cytotoxic effects in Caco-2 and HT29-MTX cell lines. In *Caenorhabditis elegans*, where insulin decreases fat storage, I-NP-PEG induced a higher reduction in the fat content than I-NP and slightly lower than the control (Orlistat). In diabetic rats, nanoparticles induced a potent hypoglycemic effect and achieved an oral bioavailability of 4.2% for I-NP and 10.2% for I-NP-PEG. This superior effect observed for I-NP-PEG would be related to their capability to diffuse through the mucus layer and reach the surface of enterocytes (where insulin would be released), whereas the mucoadhesive I-NP would remain trapped in the mucus, far away from the absorptive epithelium. In summary, PEG-coated zein nanoparticles may be an interesting device for the effective delivery of proteins through the oral route.

## 1. Introduction

The oral route is the preferred way for drug administration due to its advantages for the patient but also for the manufacturer. For the patients, this route facilitates the administration of the drug and avoids the association of the medication with a painful act (as sometimes occur with parenteral administrations); increasing the compliance of the treatment [1,2]. For the manufacturer, medications intended for the oral route have less demanding cleaning and sterility requirements than for parenteral drugs. However, the large majority of peptides and therapeutic proteins are formulated and administered as injections due to their very low bioavailability when orally administered (usually lower than 1%) [3]. The reason for this low availability is the presence of several obstacles that hamper their access to the absorptive epithelium. These obstacles include the chemical and enzymatic barriers, as well as the protective mucus layer. The chemical barrier comprises the varying pH conditions all along the gastrointestinal tract, shifting from highly acidic in the stomach (pH ≈ 1.2) to slightly basic in the colon (pH ≈ 7.5). Under these conditions, many proteins undergo a loss of their activity due to hydrolysis, deamination, or pH-induced oxidation processes [3,4]. The enzymatic barrier refers to the efficient degrading enzymes present in the lumen of the gastrointestinal tract and in the apical membrane of the enterocytes, including pepsin, trypsin, chymotrypsin, elastase, carboxypeptidases, aminopeptidases, etc. [5]. Those proteases degrade around 94–98% of all the proteins administered orally [6]. Finally, the mucus layer is a hydrogel matrix that covers the whole gastrointestinal tract and acts as the first line of defense against epithelial damage by physical, chemical, or biological aggression. This protective layer is mainly composed of mucins (highly O-glycosylated proteins), which confer a negative charge to the gel [7] and expose hydrophobic domains where proteins may bind and get retained [8].

However, the intestinal epithelium itself acts as a selective fence, hindering the pass of molecules according to their molecular weight and chemical properties. The transport of macromolecules from the lumen to the bloodstream can occur through two different pathways: transcellular or paracellular [9]. The paracellular transport occurs by passive diffusion through the spaces between two adjacent cells, spaces occupied by tight junctions that bind both cells together and act as a selective barrier [10]. However, peptides absorbed through this route must be hydrophilic, neutral, and with a low molecular weight [11]. Thus, molecules larger than 700 Da would enter the cell via active transport [9]. The transcellular pathway (a type of active transport) involves the endocytic uptake at the apical site, intracellular vesicle traffic, and basolateral efflux of the content [12]. This phenomenon occurs both in the enterocytes and in the M cells of Peyer patches localized in the intestinal epithelium [13]. Hence, due to the large molecular weight of insulin (5.8 kDa) [14], it would not be suitable to be absorbed through the paracellular route. Moreover, previous studies suggest that the transcellular transport through the enterocytes would be the main route for insulin absorption [15]. In fact, enterocytes present insulin receptors in both their apical and basolateral membranes [16]. After binding to its membrane receptor, insulin would be internalized by endocytosis and, once in the intracellular space, released to the cytoplasm [16,17].

To minimize the negative effect of these drawbacks on the bioavailability of orally administered peptides and proteins, several strategies have been proposed. These strategies include the use of polymer-drug conjugates [18,19], liposomes [20,21], SNEDDS [22,23], and polymeric nanoparticles [24,25,26]. Thus, in recent years, clinical trials with oral peptide and protein-based products have progressively increased. As a result of these research efforts, in 2019, the US-FDA approved semaglutide (Rybelsus^®^) as an oral medication against type 2 diabetes mellitus. Nevertheless, the possibility of using the oral route as a way for the administration of peptides and proteins requires the development of delivery systems adapted to the particularities of both the gut and the macromolecules.

In this context, a possible alternative may be the use of zein-based nanoparticles. Zein is a natural GRAS (Generally Recognized As Safe by the FDA) protein found in corn that has been widely used in the pharmaceutical industry, as well as for the development of polymeric nanocarriers, due to its safety, biodegradability, and low toxicity [27]. Zein nanoparticles have already demonstrated an important potential to increase the oral bioavailability of both small [28] and large molecules [29]. Moreover, although zein nanoparticles display a mucoadhesive behavior [30], their surface properties can be easily modified with hydrophilic polymers (e.g., poly(ethylene glycol)); abolishing the possibility of developing adhesive interactions with the components of the mucus layer and, thus, facilitating the possibility of reaching the intestinal epithelium. Hence, the coating of zein nanoparticles with poly(ethylene glycol) 35,000 (PEG) permits to increase their mucus diffusivity [31], which could lead to an increased oral bioavailability of the loaded biologically active compound.

The objective of this work was to evaluate the capability of two different zein-based nanoparticles, with either mucoadhesive or mucus-permeating properties, for oral protein delivery, using insulin as a model. For this purpose, the capability of both formulations to reduce the fat accumulation in *C. elegans* (regulated by insulin pathways), as well as their effect in an animal model of diabetic rats was assessed.

## 2. Materials and Methods

### 2.1. Materials

Zein, lysine, human recombinant insulin, Rose Bengal sodium salt, poly(ethylene glycol) 35,000, sodium hydroxide, Nile red, Orlistat, glucose, isopropanol, Triton X-100, agarose, trypan blue, 3-(4,5-Dimethylthiazol-2-yl)-2,5-diphenyltetrazolium bromide (MTT), dimethyl sulfoxide, trypsin-EDTA, and monobasic sodium phosphate were purchased from Sigma-Aldrich (St. Louis, MO, USA). Absolute ethanol was obtained from Scharlab (Sentmenat, Spain). 10% heat-inactivated FBS was obtained from Biochrom (Berlin, Germany). 1% penicillin/streptomycin 100X was from Biowest (Riverside, MO, USA). NEAA (non-essential amino acids) 100X were purchased from Gibco (Amarillo, TX, USA). DMEM with Ultraglutamin was from Lonza (Basilea, Switzerland). Human colon adenocarcinoma cells (Caco-2) were purchased from American Type Culture Collection (ATCC, United States). Mucus-producing cells (HT29-MTX) were kindly provided by Dr. T. Lesuffleur (INSERM U178, Villejuif, France). Isoflurane (IsoVet^®^) was from Braun (Kronberg, Germany). Sodium chloride, trifluoroacetic acid (TFA) and acetonitrile (HPLC grade) were from Merck (Darmstadt, Germany). Lumogen^®^ F Red was provided by BASF (Ludwigshafen, Germany). European bacteriological agar and peptone were acquired from Laboratorios Conda (Madrid, Spain). 96-well plates and T75 flasks were purchased from Corning Inc., (Steuben County, NY, USA). “Insulin Enzyme Immunoassay Kit” was purchased from Arbor Assays (Ann Arbor, MI, USA).

### 2.2. Preparation of Nanoparticles

#### 2.2.1. Preparation of Bare Nanoparticles Loaded with Insulin (I-NP)

A desolvation procedure previously described [31] was followed to prepare zein nanoparticles, with minor modifications. Briefly, 20 mL of a hydroalcoholic solution (61% pure ethanol in water) was used to dissolve 200 mg zein and 30 mg lysine, under magnetic stirring during 10 min. In parallel, 20 mg insulin was dissolved in 2 mL of slightly acidulated water (HCl 10 mM). Both solutions were mixed and, after 10 min of incubation, the desolvation step of zein was induced by the addition of 20 mL purified water. Afterwards, ethanol was removed using a rotatory evaporator (Büchi Rotavapor R-144; Büchi, Postfach, Switzerland) and the suspension of nanoparticles was purified by tangential flow filtration using a membrane with a molecular weight cutoff of 500 kDa (Repligen, Rancho Dominguez, CA, USA). Finally, the suspension of nanoparticles was dried in a Büchi Mini Spray Dryer B-290 apparatus (Büchi Labortechnik AG, Switzerland). The drying conditions in the spray-dryer were as follows: (i) inlet temperature, 90 °C; (ii) outlet temperature, 45–50 °C; (iii) air pressure, 4–6 bar; (iv) pumping rate, 5 mL/min; (v) aspirator, 80%; and (vi) airflow, 400–500 L/h.

Empty nanoparticles (NP), used as controls for some experiments, were prepared in the same way but in the absence of insulin.

#### 2.2.2. Preparation of Insulin-Loaded PEG-Coated Nanoparticles (I-NP-PEG)

The coating of the nanoparticles with poly(ethylene glycol) 35,000 was achieved by incubation of the just-formed nanoparticles, prior to the purification step. For this purpose, 100 mg/mL stock solution of PEG in water was prepared, and 1 mL of this stock was added to the suspension of nanoparticles. The mixture was incubated for 30 min under magnetic stirring at room temperature. Then, nanoparticles were purified and dried as described above.

Control PEG-coated nanoparticles (NP-PEG), in absence of insulin, were prepared in the same way as described above.

#### 2.2.3. Preparation of Nanoparticles Fluorescently Labeled with Lumogen^®^ F Red 305

For the fluorescently labeling of nanoparticles, 2.6 mL of a solution of Lumogen^®^ Red (0.4 mg/mL in ethanol) were added to the hydroalcoholic solution of zein and lysin and the mixture was magnetically agitated during 10 min at RT. Then, the desolvation and drying steps were conducted as aforementioned.

### 2.3. Characterization of the Physico-Chemical Properties of the Resulting Nanoparticles

#### 2.3.1. Mean Size, PDI (Polydispersity Index), ζ-Potential, and Total Process Yield

The particle size, PDI and zeta potential of the resulting formulations were assessed in a Zetasizer analyzer system (Brookhaven Instruments Corporation, Holtsville, NY, USA), after dispersion of the dried powder of nanoparticles in water. The yield of the complete process was calculated by gravimetry [32].

#### 2.3.2. Morphology Evaluation

The surface morphology and the shape of the nanoparticles were examined by scanning electron microscopy (SEM). 1.5 mg of dried nanoparticles were dispersed in 1 mL ultrapure water, mounted on SEM grids and allowed to dry overnight at room temperature. Then, the samples were coated with a gold layer using a Quorum Technologies Q150R S sputter-coated (Puslinch, ON, Canada) and images were obtained using a ZEISS Sigma 500 VP FE-SEM apparatus.

#### 2.3.3. Assessment of the Surface Hydrophobicity of the Nanoparticles

The evaluation of the surface hydrophobicity of the resulting nanoparticles was performed following the Rose Bengal method [33] with minor modifications. In brief, 500 μL-aliquots of nanoparticle suspensions at different concentrations (ranging from 0.03 to 3 mg/mL) were mixed with 1 mL of an aqueous solution of Rose Bengal dye (100 μg/mL). Then, the mixtures were incubated for 30 min at 25 °C and 1500 rpm shaking in a Labnet VorTemp 56 EVC (Labnet International Inc., Edison, NJ, USA). Afterwards, samples were centrifuged for 30 min at 13,500× *g* in a centrifuge MIKRO 220 (Hettich, Germany) and the supernatants were collected. The amount of Rose Bengal (Rose Bengal unbound) was calculated by the measurement of the absorbance at 548 nm, using a PowerWave XS Microplate reader (BioTek Instruments Inc., Winooski, VT, USA). The calculations carried out for the determination of the surface hydrophobicity are as follows:

The total surface area (TSA) of the nanoparticles was calculated assuming that the nanoparticles were completely spherical in shape and monodisperse, whose diameter is equal to the mean size determined by DLS.
TSA = (SANP) × (NTNP) (1)

In which SANP corresponds to the surface area of a single nanoparticle (4πr^2^), and NTNP represents the total number of nanoparticles present in the dilution, calculated with Equation (2):NTNP = mNP/(ρzein × VNP)(2)
where mNP is the mass of nanoparticles present in the dilution, ρzein corresponds to the density of zein (1.41 g/mL) calculated by pycnometry [34] and VNP represents the volume of a single nanoparticle, calculated using as diameter value the mean size obtained in the Zetasizer.

The hydrophobicity of the formulation was determined by the slope of the line obtained by the representation of the TSA (X axis) against the partitioning quotient (PQ) (Y axis), where PQ is the quotient between the amount of Rose Bengal bound and unbound. The higher the slope of the line, the higher the hydrophobicity of the formulation.

#### 2.3.4. Nanoparticles Surface Evaluation by Fourier Transform Infrared Resonance (FTIR)

The Fourier transform infrared spectroscopy (FTIR) spectra of the nanoparticles were obtained using a Fourier transform spectrophotometer IR Affinity-1S (Shimadzu, Kyoto, Japan) coupled to a Specac Golden Gate ATR. For that purpose, a small amount of powder of nanoparticles was placed over the diamond and the reflectance spectra were obtained scanning from 600 to 4000 cm^−1^ at 2 cm^−1^ of resolution, and 50 scans per spectrum. Then, the Labsolution IR software was used to analyze the spectra.

### 2.4. Insulin Analysis

#### 2.4.1. Insulin Payload and Encapsulation Efficiency

The insulin loading of the resulting nanoparticles was quantified by ELISA. For this purpose, nanoparticles were dissolved in ethanol 70% and diluted with purified water until a theoretical concentration of 600 pg/mL insulin. Afterwards, the quantification was carried out following the specifications described by the manufacturer. The payload was expressed as the amount of insulin (μg) per milligram of nanoparticles and, the encapsulation efficiency (EE, expressed as a percentage), was calculated as the quotient between the amount of insulin quantified and the total amount of insulin added for the formulation of the nanoparticles.

#### 2.4.2. In Vitro Release Studies

Insulin release studies from the nanoparticles were performed in simulated gastric fluid (SGF; pH 1.2) and simulated intestinal fluid (SIF; pH 6.8). For this purpose, 5 mL of aqueous suspensions of nanoparticles (10 mg/mL) were placed into Float-A-lyzer^®^ devices with a molecular weight cutoff of 300 kDa (Spectrum Labs, Breda, The Netherlands). Then, the devices were placed in vessels containing 45 mL SGF and kept for 2 h under magnetic agitation. After 2 h of incubation in SGF, the devices were transferred to other vessels containing 45 mL SIF and incubated for 24 h. The whole experiment was carried out at 37 °C. At determined times, 200 μL were withdrawn and replaced with 200 μL fresh medium. Samples were analyzed by HPLC.

The amount of insulin was quantified in an Agilent model 1200 series (Agilent Technologies, Waldbronn, Germany), coupled with a photodiode array detection system at 240 nm. A Jupiter column (5 μm C18 300 A, 150 × 4.6 mm; Phenomenex, CA, USA) was used as stationary phase, whereas the mobile phase was composed of an isocratic mixture of acetonitrile and 0.1% trifluoroacetic acid in water (7:3 by vol.). The flow rate was 0.8 mL/min and the temperature of the column was set to 25 °C. The calibration curve was performed using 8 different insulin solutions with concentrations ranging from 7 to 1000 µg/mL (1000, 500, 250, 125, 62.5, 31.25, 15.62 and 7.8 µg/mL). Good linearity (R^2^ = 0.9994) was obtained in the studied range. Under these conditions, the detection limit for insulin quantification was found to be 7 µg/mL.

### 2.5. Cellular Studies

#### 2.5.1. Caco-2 Cell Culture

The human colon carcinoma cell line Caco-2 was used as a model of the intestinal epithelial cell due to its property of forming a monolayer of polarized cells with the phenotype of the small intestine [35]. Moreover, epithelial cells represent the most abundant type of cell in the gastrointestinal tract. Caco-2 cells were incubated at 37 °C in a 5% CO_2_/95% O_2_ atmosphere and 90% relative humidity in an incubator (ESCO CelCulture^®^ CO_2_ incubator, Singapore). Cells were cultured in Dulbecco’s Modified Eagle Medium with Ultraglutamine (DMEM ATCC) supplemented with 10% heat-inactivated fetal bovine serum (FBS), 1% (*v*/*v*) penicillin/streptomycin (100 U/mL and 100 µg/mL, respectively) and 1% (*v*/*v*) non-essential amino acids (NEAA). Cell passages were performed once a week with trypsin-EDTA (0.25%, 0.05%) and seeded in 75 cm^2^ flasks at a density of 4 × 10^5^ cells/flask. The medium was changed every two days.

#### 2.5.2. HT29-MTX Cell Culture

The mucus-secreting HT29-MTX cell line was used as model of intestinal goblet cells due to their phenotypical similarities. In addition, mucus-secreting cells correspond to the second most abundant population of cells in the gastrointestinal tract [36]. HT29-MTX cells were maintained at 37 °C in a 5% CO_2_/95% O_2_ atmosphere and 90% relative humidity in an incubator (ESCO CelCulture^®^ CO_2_ incubator, Singapore). Cells were cultured in Dulbecco’s Modified Eagle Medium (DMEM) supplemented with 10% FBS, 1% penicillin/streptomycin and 1% NEAA. The medium was changed every two days and cells were subcultured once a week with trypsin-EDTA at a density of 4 × 10^5^ cells/75 cm^2^ flask.

#### 2.5.3. Cytotoxicity Evaluation

The potential cytotoxic effect of insulin-loaded nanoparticles was assessed against Caco-2 (passage 70–75) and HT29-MTX (passage 45–50) cell lines using the MTT procedure. Both cell lines were separately grown in culture flasks with the complete medium, as described before. Cells were detached from the flask, seeded into 96-well plates at 20,000 cells/well for Caco-2 and 10,000 for HT29-MTX cell lines, and incubated for 24 h. Afterwards, the medium was removed, and cells were washed twice with phosphate-buffered saline (PBS) prior to the addition of different concentrations (1.7–70 µg/mL) of insulin loaded into nanoparticles dispersed in fresh medium without FBS supplementation. A positive control without nanoparticles, and a negative control of Triton X-100 (1%) were also added. After 24 h of incubation, the media were removed, and cells were washed twice with PBS. Then, the MTT solution (0.5 mg/mL in cell-culture medium) was added to each well and incubated for 4 h in a dark environment. Subsequently, the MTT solution was removed and 200 µL DMSO was added to each well in order to solubilize formazan crystals. Plates with DMSO were shaken in an orbital shaker for 15 min at room temperature and covered from light. Then, the absorbances were measured in a plate reader (Biotek Synergy 2, Winooski, VT, USA) at 570 and 630 nm wavelength. The cell viability was calculated according to the following equation:(3)$$\mathrm{Cell}\;\mathrm{viability}\;\left(\%\right)=\frac{\mathrm{sample}\;\mathrm{value}\;-\mathrm{negative}\;\mathrm{control}}{\mathrm{positive}\;\mathrm{control}-\mathrm{negative}\;\mathrm{control}\;}\times 100$$

### 2.6. In Vivo Evaluation of Insulin-Loaded Nanoparticles in an Animal Model of Caenorhabditis elegans

#### 2.6.1. Strain and Culture Conditions

*Caernoharbditis elegans* (*C. elegans*) was chosen as an animal model for a first-step evaluation of the suitability of insulin-loaded zein nanoparticles. *C. elegans* is a nematode sensitive to human insulin in which the hormone binds to the DAF-2 receptor and modulates its downstream signaling cascade, including fat accumulation [34,37]. For this purpose, C. elegans were maintained and cultured as described previously [37]. The Wild-type N2 Bristol strain was obtained from the Caenorhabditis Genetics Center (CGC, University of Minnesota, MN), and was cultured at 20 °C on NGM (Nematode Growth Medium) agar, with *Escherichia coli* OP50 as a normal nematode diet. For all experiments, age-synchronized worms were obtained by bleaching with hypochlorite, a condition in which only eggs can survive. Recovered eggs were left to hatch overnight in an M9 buffer solution.

#### 2.6.2. Nanoparticles Intake

To evaluate the intake of nanoparticles by the worms, Lumogen^®^ red-loaded nanoparticles were supplemented to the growth medium of *C. elegans*. After the preparation of the NGM plates, a suspension of fluorescent nanoparticles was added and let dry. Afterwards, 50 µL of a culture of OP50 were added over the solid NGM plates supplemented with the nanoparticles and let dry in darkness overnight. About 500 adult (L4 stage) worms were placed onto each plate and incubated for 2 h at 20 °C. After the incubation time, worms were collected and fixed with agarose over glass slides. Agarose was prepared at a 2% concentration (*w*/*v*) and supplemented with 1% sodium azide to kill the worms during fixation. Glass slides containing the fixed worms were observed with a Nikon eclipse 80i epi-fluorescent microscope. Two different filters were used to observe the glass slides: the rhodamine filter, to visualize the Lumogen^®^-loaded nanoparticles, and the DAPI filter to observe the whole body of the worms (*C. elegans* shows autofluorescence under the DAPI filter).

#### 2.6.3. In Vivo Efficacy of the Nanoparticles in *C. elegans*

The evaluation of the efficacy of insulin-loaded nanoparticles was performed by the quantification of the fat tissue inside the worm, as described previously [38]. The assays were carried out in triplicates in 6-well plates containing 4 mL glucose-supplemented NGM (0.5% *w*/*v*) per well. The insulin treatment (free or nanoencapsulated) was added at a concentration of 50 µg/mL in NGM. Treatments of empty nanoparticles (NP and NP-PEG) were also added to evaluate the effect of the vehicle itself. A positive control of Orlistat (6 µg/mL), as a fat-reducing agent, was employed.

L1 larvae worms were transferred into the wells containing the NGM supplemented with treatments and were allowed to grow until the L4 larvae stage (approximately 46 h). Afterwards, worms were harvested and washed with 0.01% triton-X in phosphate-buffered saline (PBST). Washed worms were fixed with 40% isopropanol and stained by the addition of Nile Red solution (3 µg/mL) and incubation for 30 min at room temperature and with soft rocking. Finally, stained worms were fixed in 2% agarose over glass slides. Images of the worms were obtained using a Nikon’s SMZ18 stereomicroscope (Nikon Instruments Inc., Japan) attached to a DS-FI1C refrigerated color digital camera and an epi-fluorescence system. For the acquisition of the images, a GFP filter was used (Ex 480–500; DM 505; BA 535–550). The post-acquisition processing of the images and the fat quantification were carried out using the FIJI (image J) software.

### 2.7. In Vivo Evaluation of Insulin-Loaded Nanoparticles in Diabetic Rats

#### 2.7.1. Strain and Housing Conditions

Healthy male Wistar rats were purchased from Envigo (Indianapolis, USA) with a weight in the range of 180–220 g. They were housed with 12-h dark/light cycles under controlled temperature (23 ± 2 °C) and with free access to food and water. Upon arrival, animals were allowed to acclimate at least for one week before any manipulation. During the procedures, animals had free access to water but were deprived of food. All the manipulations were carried out following an approved protocol by the “Ethical and Biosafety Committee for Research on Animals” from the University of Navarra, following the European legislation on animal experimentation (protocol 063-20).

#### 2.7.2. Induction of Diabetes

Diabetes was induced by a single intraperitoneal injection of streptozotocin (80 mg/kg) to overnight fasted animals [29]. For that purpose, streptozotocin was dissolved in 0.1 M citrate buffer (pH 4.5). After 5–8 days, rats with frequent urination and fasting blood glucose levels higher than 250 mg/dL were considered as diabetic and, thus, randomized for further studies.

#### 2.7.3. Efficacy Evaluation

The efficacy evaluation was carried out in diabetic rats fasted for 12 h prior to any administration. All the administrations were performed through oral gavage by using a stainless-steel cannula, except otherwise stated. Animals were divided into the following four groups (*n* = 6): (i) control animals receiving 1 mL purified water; (ii) I-NP group, receiving a suspension of bare nanoparticles loaded with insulin, dispersed in purified water, at a dose of 50 IU/kg; (iii) I-NP-PEG group, receiving a suspension of insulin-loaded PEG-coated nanoparticles, dispersed in purified water, at a dose of 50 IU/kg; (iv) Ins sc group, receiving a subcutaneous administration of an aqueous solution of insulin at a dose of 5 IU/kg.

Blood samples were collected from the tail vein in isoflurane-anesthetized animals. Samples were collected prior to the administration of the treatments, to establish the glucose baselines, and at fixed times after the administration. At every extraction point, blood glucose levels were analyzed with an Accu-Check^®^ Aviva glucometer (Roche Diagnostics, Basel, Switzerland). Insulin levels in blood were quantified using an ELISA kit.

#### 2.7.4. Pharmacokinetic and Pharmacodynamic Analysis

The hypoglycemic effect was estimated by calculating the area above the curve (AAC_0–6h_) from the blood glucose curves. The calculation was carried out by using the trapezoidal method, calculated using the PKsolver software [39]. The relative pharmacological availability (PA) was calculated as the percentage of the hypoglycemic effect induced by the formulations compared to the sc insulin (Equation (3)) AACoral
(4)$$\mathrm{PA}=\frac{\mathrm{AAC}\;\mathrm{oral}\;\times \;\mathrm{Dose}\;\mathrm{sc}}{\mathrm{AAC}\;\mathrm{sc}\;\times \;\mathrm{Dose}\;\mathrm{oral}}\times 100$$

The main pharmacokinetic parameters (C_max_, T_max_ and area under the curve (AUC_0–6h_)) were calculated from the representation of the serum insulin levels vs time, again using the PKsolver software. The relative bioavailability of oral formulations (Fr) was calculated as the percentage of AUC of oral formulations compared to the subcutaneous injection.

### 2.8. Statistical Analysis

For statistical analyses, the means and standard errors of each data set were calculated. The comparisons between groups were carried out using a one-way ANOVA test followed by a multiple comparison test (Tukey–Kramer test), except for pharmacokinetic and pharmacodynamic studies in which t-student was used to compare I-NP to I-NP-PEG. Significant differences are marked as follows: * *p* < 0.05, ** *p* < 0.01 or *** *p* < 0.001. All calculations were performed using GraphPad Prism v6 (GraphPad Software, San Diego, CA, USA) and the curves were plotted with the Origin 8 software (OriginLab Corp, Northampton, MA, USA).

## 3. Results

### 3.1. Characterization of Nanoparticles

Table 1 summarizes the main physicochemical properties of empty and insulin-loaded nanoparticles. Zein nanoparticles, containing insulin, were obtained by desolvation and, eventually, “decorated” by incubation with PEG 35,000. Then, the nanoparticles were purified and dried. The encapsulation of insulin increased the mean size of the resulting nanoparticles (from about 230 nm to 270 nm) and decreased the negative zeta potential (from −54 mV to −39 mV). In a similar way, the coating of nanoparticles with PEG 35,000 did not importantly affect the size or the zeta potential of the resulting nanoparticles, compared to bare ones (Table 1). The insulin loading was calculated to be close to 80 µg/mg nanoparticles. Again, the coating of nanoparticles with PEG 35,000 did not modify their insulin payload.

The surface morphology and shape of insulin-loaded nanoparticles, evaluated by SEM, are shown in Figure 1. Both types of nanoparticles were spherical in shape but with differences in surface morphology. While bare nanoparticles (I-NP) showed a rough surface, PEG-coated nanoparticles showed a smooth surface, without perceptible irregularities.

The surface hydrophobicity of the nanoparticles, calculated by the Rose Bengal test, is shown in Figure 2. Interestingly, the encapsulation of insulin in bare nanoparticles increased the hydrophobicity of the resulting nanoparticles (Figure 2; *p* < 0.05). On the contrary, the incorporation of insulin in PEG-coated nanoparticles did not significantly modify their hydrophobicity, when compared to empty NP-PEG.

Figure 3 shows the FTIR spectra of the different formulations, as well as the raw materials employed for the formulation of the nanoparticles. Both proteins (zein and insulin) displayed characteristic absorption bands corresponding to amide I, and II groups. In free zein, as well as in zein nanoparticles, a band centered at about 1647 cm^−1^ (mainly attributed to the C=O stretching vibration of amide I) was detected. In addition, the –N-H bending coupled to –C-N stretching vibration (amide II) was observed as a broad band at 1517cm^−1^. However, in the insulin spectrum, both signals appeared as multi- shouldered broad bands centered at 1639 cm^−1^ for amide I and 1512 cm^−1^ for amide II. Regarding the amide III (combination of N-H in-plane bending and C-N vibrations), also characteristic of proteins, was observed as multiple signals (1247–1352 cm^−1^) in the spectrum of zein nanoparticles. For insulin, the amide II band appeared as a broad band centered at 1236 cm^−1^. It is worth noting that the spectrum of I-NP clearly showed the characteristic vibration band corresponding to insulin amide I (1639 cm^−1^) together with the superposition of signals corresponding to insulin and zein amide II. Moreover, the appearance of multiple signals in the amide III vibration region was associated with an overlapping of signals from both proteins (insulin and zein). The slight displacement of insulin vibration bands of 1236 to 1240 cm^−1^ (among others) could be attributed to a weak interaction between insulin and zein as a result of the hormone encapsulation. In the spectrum of I-NP-PEG, insulin bands previously mentioned (mainly amide I and II) together with multiple PEG signals (1465, 1338, 1276, 1240, 1145, 1101 cm^−1^) and those corresponding to the polymer fingerprint region (1060, 960 and 840 cm^−1^) were presented. In addition, the displacement of some of them (e.g., 1093 cm^−1^ to 1101 cm^−1^ corresponding to PEG alcoholic group) confirmed the interaction between zein and PEG.

### 3.2. In Vitro Release Behavior of Insulin-Loaded Nanoparticles

The release behavior of the formulations during their incubation in SGF and SIF is shown in Figure 4. Both formulations (I-NP and I-NP-PEG) displayed a similar release profile; although, the coating of zein nanoparticles with PEG 35,000 induced a slightly slower release rate, compared to bare nanoparticles. In SGF, after 2 h of incubation, the amount of insulin released from I-NP represented about 37% of the payload. For I-NP-PEG, only 30% of the insulin loading was released in 2 h of incubation. When nanoparticles were transferred to the SIF, the release rate for both formulations appeared to be similar to that observed in SGF, evidencing the null effect of the pH on the release of insulin from the nanoparticles. After 22h in SIF, both formulations of nanoparticles had released the total content of the initial payload.

### 3.3. Cytotoxicity Evaluation of Insulin-Loaded Nanoparticles

The cytotoxic effect of insulin-loaded nanoparticles (I-NP and I-NP-PEG) over two cell lines that mimic the most predominant cell types in the intestines, Caco-2 cells (epithelial cells) and HT29-MTX cells (mucus-secreting cells), were quantified by the MTT assay. Figure 5 displays the results of the cell viability after 24 h of incubation of the cells with either bare or PEG-coated insulin-loaded nanoparticles. After 24 h of incubation of the cell lines with the treatments, none of the formulations showed a cytotoxic effect at any of the concentrations tested, ranging from 1.7 to 70 µg/mL of insulin.

### 3.4. Effect of Insulin-Loaded Nanoparticles in C. elegans

The effect of insulin (free or loaded in zein nanoparticles) on the fat content of *C. elegans* grown in a medium (NGM) supplemented with glucose is presented in Figure 6. As expected, the incorporation of free insulin (50 µg/mL) to the medium induced a reduction in the fat accumulated by the worms (*p* < 0.01). This reduction was calculated to be about 15%, in comparison with control animals.

Regarding the effect of nanoparticles, the first step was to confirm if the worms ate the nanoparticles. As presented in Figure 7, the fluorescence caused by the presence of lumogen red in nanoparticles (incorporated in the medium in which the worms were grown) is observed all along the gastrointestinal tract of the animal. For I-NP-treated worms, the amount of fat accumulated in these animals was calculated to be about 30% lower than for control worms (NGM group in Figure 6). In a similar way, animals treated with I-NP-PEG displayed a reduction in their fat content close to 40%; slightly lower than the effect produced by Orlistat (6 µg/mL; employed as positive control), which was quantified to be 42%. Another interesting fact was to evidence that empty zein nanoparticles (both bare and PEG-coated) also induced a decrease in the fat content of the worms (about 11% for NP and 14% for NP-PEG).

### 3.5. Evaluation of the Hypoglycemic Activity in Diabetic Rats

Figure 8 shows the evolution of glycemia in animals as a function of the treatment. Animals receiving a subcutaneous injection of insulin solution (5 IU/kg) displayed a rapid decrease in blood glucose levels. Thus, 2 h post-administration, the glycemia of these animals represented 22% of the initial blood glucose levels, reaching blood glucose levels below 120 mg/dL. These low blood glucose values were maintained for at least four more hours. On the other hand, animals treated with insulin-loaded nanoparticles exhibited a slower and more sustained decrease in glycemia. Hence, one hour after the administration of I-NP and I-NP-PEG, both groups of animals presented similar values of glucose in the blood. For later extraction points, on the one hand, animals treated with I-NP showed the same glycemic values without any relevant further reduction. This group of animals achieved a maximum decrease in the blood glucose levels of 57%, compared to initial values, displaying glycemic values close to 280 mg/dL. On the other hand, rats treated orally with I-NP-PEG exposed decreasing values of glycemia with time, reaching the highest decrease 6 h after the administration, with 32% of the initial values, corresponding to blood glucose levels around 170 mg/dL. Moreover, at this extraction point, the glycemic values of animals treated with I-NP-PEG matched the values of those treated with the subcutaneous injection of insulin.

From the curves of glycemia over time, the main pharmacodynamic parameters were calculated (Table 2). The most remarkable finding is that the oral administration of I-NP-PEG decreased the blood glucose levels (C_min_) to almost the same values as the subcutaneous injection of insulin. Furthermore, this formulation needed 2 h more to induce the maximum decrease in glycemia (T_max_) than for animals treated with either subcutaneous insulin or with oral nanoparticles (5 h vs. 3 h, respectively). It is also important to highlight that the coating of the nanoparticles with PEG increased more than 3-fold the pharmacological availability (PA) of insulin administered in bare nanoparticles. The PA of I-NP was calculated to be 4.7% compared to the control (subcutaneous insulin) while, for I-NP-PEG, the PA was calculated to be close to 15%.

Figure 9 shows the plasma levels of human insulin in rats receiving a subcutaneous administration of insulin (Ins sc; 5 IU/kg) or an oral administration of nanoencapsulated insulin (I-NP or I-NP-PEG; 50 IU/kg). The main pharmacodynamic parameters, calculated from the insulin plasma levels, are summarized in Table 3. Animals treated with a subcutaneous dose of insulin showed the typical profile characterized by a fast rise, reaching the C_max_ during the first 2 h post-administration, achieving insulinemic values of almost 6 ng/mL. Afterwards, plasma levels of insulin decreased to almost undetectable levels after 4 h. For oral nanoencapsulated insulin, both formulations showed the same pattern, characterized by a rapid increase in the blood levels of insulin 1 h post-administration, when the insulin levels in blood were found to be around 1 ng/mL for I-NP and 2.5 ng/mL for I-NP-PEG. The fast rise observed 1 h after the oral administration of I-NP and I-NP-PEG was followed by a plateau state that lasted several hours. Despite both formulations showing the same profile, PEG-coated nanoparticles produced blood levels of insulin which were about two-fold than those obtained with bare nanoparticles. As a result, the relative oral bioavailability of insulin formulated in bare nanoparticles was calculated to be 4.2%, whereas, for I-NP-PEG, this value increased almost 2.5 times (10.2%).

## 4. Discussion

Orally administered therapeutic proteins and peptides need to face several biological barriers that make the oral bioavailability of the drug almost negligible. In principle, the use of nanoparticles for the oral delivery of such compounds may be an adequate approach to protect the loaded cargo against premature degradation, including the acidic conditions of the stomach, the presence of enzymes, as well as the mechanical stress in the lumen (e.g., osmotic pressure and peristalsis) [40,41]. On the other hand, the design of nanoparticles with either mucoadhesive or mucus-permeating properties may be of interest to increase the residence time of the drug delivery system in close contact with the epithelium, improving the probabilities for insulin absorption and/or interaction with its receptor [17].

From a general point of view, protein-based nanoparticles offer some advantages for drug delivery purposes, including their biodegradability and capability to accommodate a high variety of compounds in a non-specific way [42]. In this context, zein possesses some peculiarities that provide additional advantages for the formulation of nanoparticles. First, an optimal regulatory status. Second, its lipophilic character that facilitate both the preparation of stable nanoparticles, without the need for stabilization or cross-linking procedures [29], and the modification of their surface with different coating agents by simple non-covalent based procedures [31,43]. Third, its lower allergenicity [44] and slower digestion than other proteins [45].

In this work, zein-based nanoparticles with either mucoadhesive (NP) or mucus-permeating properties (NP-PEG) were selected to evaluate and compare their capability to deliver insulin orally. Both types of nanoparticles (bare and PEG-coated) displayed a mean size of around 270 nm and negative zeta potential (Table 1). Noteworthy, the surface zeta potential of insulin-loaded nanoparticles was markedly less negative than empty nanoparticles, suggesting that, in the formation of insulin-loaded nanoparticles, some structural changes may happen during the insulin encapsulation. The insulin payload of the nanoparticles was calculated to be around 8%. However, the insulin release profile of PEG-coated nanoparticles has been demonstrated to be slightly slower and more sustained than for bare nanoparticles. This may be caused by the presence of the PEG coating, which generates a shell that surrounds the nanoparticle and delays the release of insulin to the outer medium. It is worth noting that the pH conditions of the SGF and SIF seemed not to affect the insulin release profiles, denoting stability of the system in both incubation media.

FTIR analysis was employed to put in evidence the presence of both insulin and PEG in their corresponding formulations. Likely, the encapsulation of inulin did not affect the coating of nanoparticles, since the displacement of the PEG bands was similar for both empty and insulin-loaded formulations (Figure 3). Zein nanoparticles do not display a shell-core structure, but a fractal-like conformation composed of 20 nm spherical blocks of zein that, during the desolvation, collide to form the nanoparticle [38]. Thus, during the formation of nanoparticles, some insulin molecules may get adhered to the surface of the nanoparticle and, hence, induce the changes in the surface hydrophobicity for I-NP observed in Figure 2.

The safety of the formulations has been evidenced by the cytotoxicity evaluation carried out in Caco-2 and HT29-MTX cells. Since our formulations are intended for oral delivery purposes, these two cell lines were selected for being widely used as a model for intestinal epithelial cells (Caco-2) [46] and intestinal goblet cells (HT29-MTX) [47]. The cell viability of both cell lines has been demonstrated not to be affected by the incubation with insulin-loaded nanoparticles, either uncoated or PEG-coated. Thus, since no toxicity was observed in these cellular models, no toxic effects would be expectable in greater animal models such as *C. elegans* or rodents receiving insulin-loaded zein nanoparticles through the oral route.

*C. elegans* is an animal with insulin signaling pathways highly conserved with humans. Under high-glucose conditions, these nematodes experience an expansion of the adipose tissue caused by the activation of the DAF-2 receptor [34]. Likely, human insulin exerts an antagonistic effect over this receptor, blocking the metabolic pathways that lead to increased fat storage [48,49]. On the other hand, the intestine of *C. elegans* has some homologies with that of mammals [50], including the presence of an absorptive layer of ciliated cells covered by a glycocalyx. These characteristics make this animal model an interesting instrument for the evaluation of insulin-loaded nanoparticles intended for oral administration. When the worms were treated with insulin, as expected, a decrease in the fat accumulation was observed (Figure 6). This decrease in the fat content of worms was significantly higher when insulin was administered into nanoparticles; particularly when insulin was formulated in PEG-coated nanoparticles. This observation may rely on the protection conferred by the nanoparticles against the physical and chemical digestion in the gastrointestinal tract of the worms. The superior capability of PEG-coated nanoparticles to decrease the accumulation of fat, when compared with bare nanoparticles, would be related to the increased diffusivity of the nanoparticles in mucus, facilitating their arrival to the epithelium surface [31].

In rats with streptozotocin-induced diabetes, the administration of a subcutaneous injection of insulin showed a fast decrease in the glycemia, reaching the lowest values at around 2–3 h post-administration. This result is in the line with previously reported [51,52,53,54]. On the other hand, administration of insulin-loaded nanoparticles induced a hypoglycemic effect that started 1 h post-administration and lasted for at least 5 h more, being I-NP-PEG significantly more potent I-NP (*p* < 0.001). This fact suggests that the capability of zein nanoparticles to effectively deliver insulin through the oral route would be directly determined by their biodistribution within the gut. While bare nanoparticles, with a mucoadhesive behavior [34], are mainly trapped in the protective mucus layer (remaining away from the absorptive epithelium), PEG-coated nanoparticles are able to diffuse through the mucus layer and reach the surface of enterocytes [31]. Moreover, PEG-coated nanoparticles have shown the capability of reaching the cecum more rapidly than bare nanoparticles and, despite the cecum function being mainly to absorb water and small molecules, insulin can also be absorbed in this section of the gut [55,56]. The arrival of nanoparticles close to the surface of the intestinal epithelium, and onsite release of the loaded hormone, would facilitate the interaction of insulin with its receptor and, thus, the absorption of the peptide through the receptor-mediated endocytosis [17]. The sustained release of insulin from the nanoparticles (as observed in Figure 4) is reflected in the sustained levels of insulin in the rats’ blood.

In our work, the relative oral bioavailability of insulin formulated in PEG-coated nanoparticles was calculated to be close to 10%, 2.5-fold higher than for bare nanoparticles. This oral bioavailability is higher than other values previously published with PLGA-chitosan composite nanocarriers (about 8%; [57]) or SNEDDS (about 1.8%; [58]); although, it is lower (about 2-times) when compared with other previous results obtained with liposomes decorated with PEG and folic acid (19.08%; [59]), folate-chitosan nanoparticles (17.04%; [60]), or zein/caseinate-based nanoparticles co-encapsulating insulin and cholic acid (20.5%; [61]). In spite of the lower capability to promote the oral absorption of insulin, our formulation offers some interesting advantages that may facilitate a translational approach; particularly in those aspects related to the scale-up of a reproducible process (including the drying step) and the simplification of non-clinical toxicity assessments of the regulatory dossier. Among others, PEG-coated zein nanoparticles may be obtained in a simple and scalable preparative process that only requires the use of pharmaceutical acceptable reagents and solvents (ethanol and water) and allows the generation of a powder formulation, easily dispersible in water, with a high insulin payload. Furthermore, as described previously [31], PEG-coated zein nanoparticles do not enter the systemic circulation minimizing a possible accumulation in the body and toxicological issues.

## 5. Conclusions

In summary, zein nanoparticles can be used to encapsulate insulin and produce (through a simple and reliable method) cheap, safe, and efficient nanocarriers for oral delivery purposes that would protect the cargo in the harsh conditions of the gut after an oral administration. The resulting nanoparticles display an appropriate size and surface zeta potential, with high entrapment efficiencies. Moreover, the coating of insulin-loaded nanoparticles with a hydrophilic polymer (PEG) did not alter the physicochemical properties of the nanoparticles and led to a significant increase in the oral bioavailability of insulin and a more potent hypoglycemic effect. Thus, this type of nanocarrier might be a suitable tool for the effective delivery of insulin through the oral route.

## Figures and Tables

**Figure 1 pharmaceutics-14-00039-f001:**
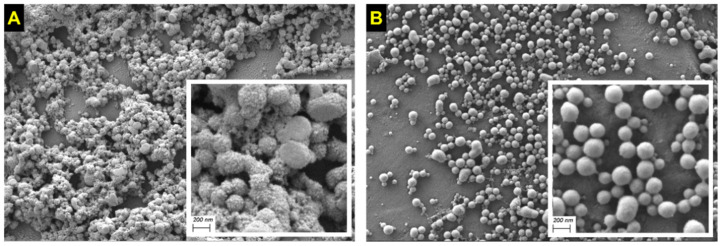
SEM microphotographs of insulin-loaded nanoparticles. (**A**) bare nanoparticles (I-NP); (**B**) PEG-coated nanoparticles (I-NP-PEG). White squares are magnifications of the nanoparticles, in which the scale bar corresponds to 200 nm.

**Figure 2 pharmaceutics-14-00039-f002:**
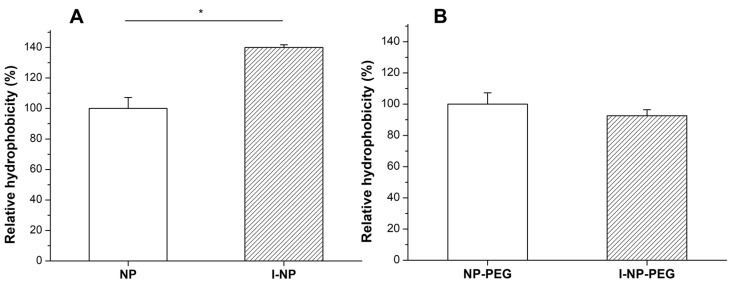
Surface hydrophobicity of (**A**) empty and insulin-loaded bare nanoparticles (NP and I-NP); (**B**) empty and insulin-loaded PEG-coated nanoparticles (NP-PEG and I-NP-PEG). Values are normalized to the hydrophobicity of the empty formulation (NP or NP-PEG). Data expressed as mean ± S.D. (*n* = 3). *: *p* < 0.05.

**Figure 3 pharmaceutics-14-00039-f003:**
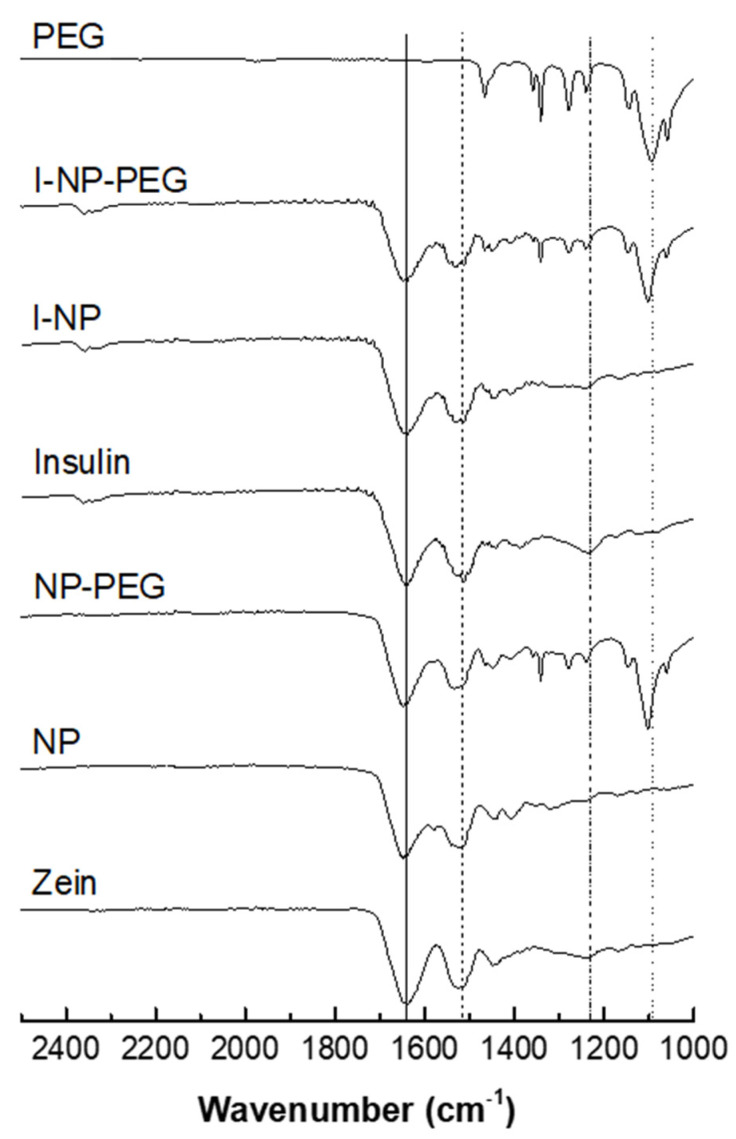
FTIR spectra of PEG 35,000, zein, insulin, empty zein nanoparticles uncoated (NP) and coated with PEG (NP-PEG), and insulin-loaded nanoparticles with (I-NP-PEG) and without PEG coating (I-NP). Straight line corresponds to the 1647 cm^−1^ band belonging to the amide I group; dashed line corresponds to the 1517 cm^−1^ band belonging to the amide II group; dashed-dotted line corresponds to the 1236 cm^−1^ band belonging to the amide III group; dotted line corresponds to the 1096 cm^−1^ band belonging to the alcoholic group of PEG.

**Figure 4 pharmaceutics-14-00039-f004:**
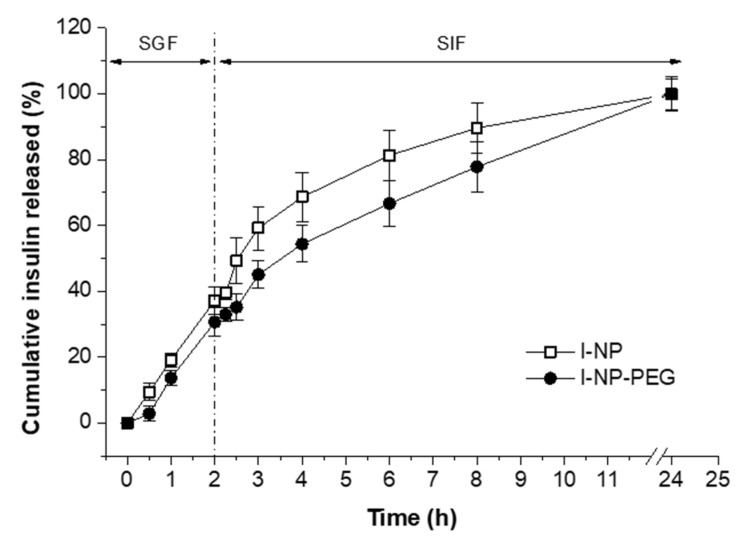
In vitro release behavior of insulin from I-NP and I-NP-PEG in SGF and SIF at 37 °C. Data presented as mean ± S.D. (*n* = 3).

**Figure 5 pharmaceutics-14-00039-f005:**
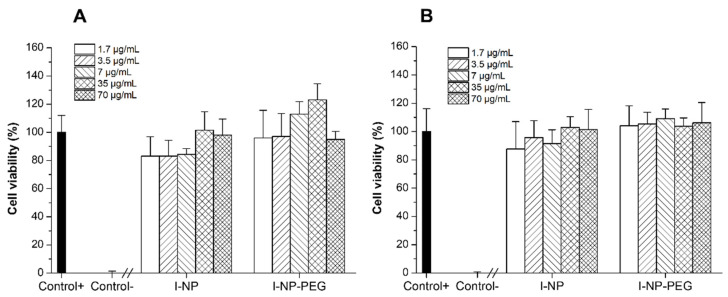
Effect of insulin-loaded nanoparticles (I-NP and I-NP-PEG) on the viability of (**A**) Caco-2 cells, and (**B**) HT29-MTX cell lines. Data presented as mean ± S.D. (*n* = 3).

**Figure 6 pharmaceutics-14-00039-f006:**
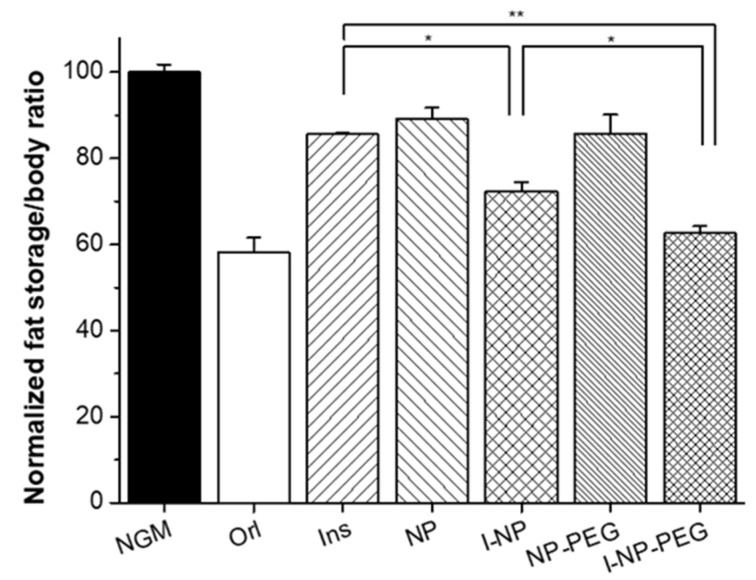
Effect over the fat storage of *C. elegans* cultured under high-glucose conditions of empty and insulin-loaded zein nanoparticles (coated with PEG and uncoated), as well as a solution of free insulin. Data presented as mean ± S.D. (*n* ≥ 75 worms). * *p* < 0.05; ** *p* < 0.01. Orl: Orlistat; Ins: free insulin; NP: empty bare zein nanoparticles; I-NP: insulin-loaded bare nanoparticles; NP-PEG: empty PEG-coated nanoparticles; I-NP-PEG: insulin-loaded PEG-coated nanoparticles.

**Figure 7 pharmaceutics-14-00039-f007:**
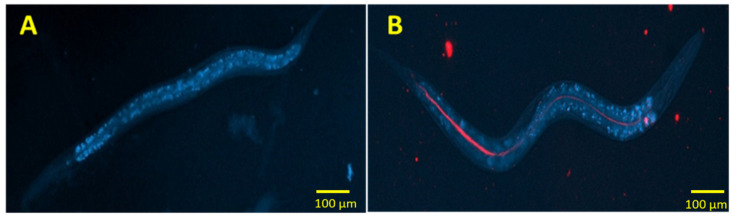
Fluorescence micrographs of *C. elegans*. (**A**) self-fluorescence of a worm under DAPI filter. (**B**) Worm fed with fluorescently-tagged nanoparticles.

**Figure 8 pharmaceutics-14-00039-f008:**
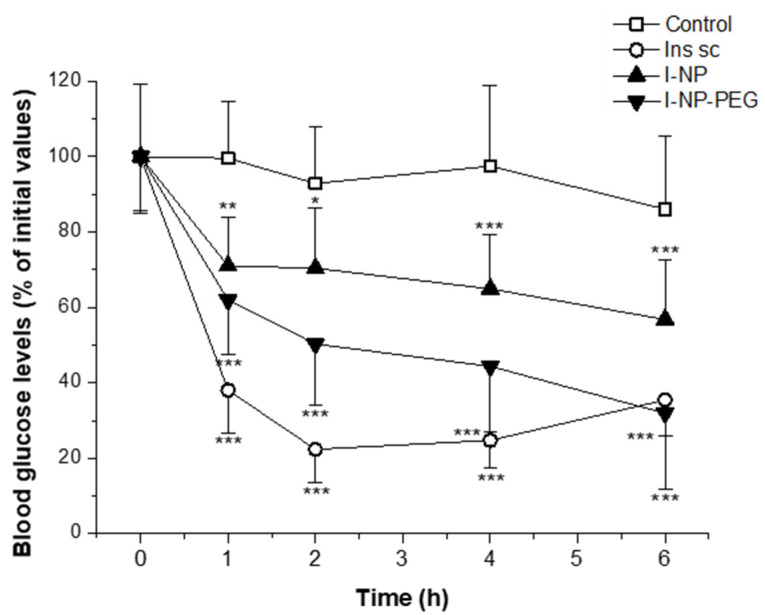
Blood glucose levels (% of initial values) of streptozotocin-induced diabetic rats after administration of subcutaneous solution of insulin (Ins sc; 5 IU/Kg) or oral administration of water (control), insulin-loaded bare nanoparticles (I-NP; 50 IU/Kg), or insulin-loaded PEG-coated nanoparticles (I-NP-PEG; 50 IU/Kg). Data expressed as mean ± S.D. (*n* ≥ 6). * *p* < 0.05; ** *p* < 0.01; *** *p* < 0.001, compared to control.

**Figure 9 pharmaceutics-14-00039-f009:**
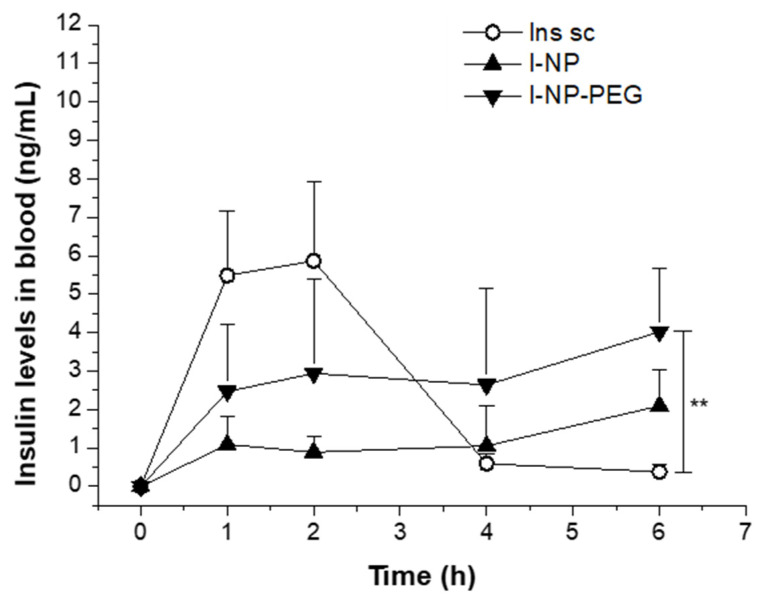
Serum insulin levels vs time after the subcutaneous administration of an insulin solution (Ins sc; 5 IU/Kg) or oral administration insulin-loaded bare nanoparticles (I-NP; 50 IU/Kg), or insulin-loaded PEG-coated nanoparticles (I-NP-PEG; 50 IU/Kg). Data expressed as mean ± S.D. (*n* ≥ 6). **: *p* < 0.01.

**Table 1 pharmaceutics-14-00039-t001:** Physico-chemical characteristics of empty (NP and NP-PEG) and insulin-loaded nanoparticles (I-NP and I-NP-PEG). Data expressed as mean ± S.D. (*n* ≥ 3).

Formulation	Size (nm)	PDI	Zeta Potential (mV)	Insulin Payload (µg/mg)	E. E. (%)
NP	239 ± 19	0.17 ± 0.07	−56.7 ± 3.4	-	-
NP-PEG	222 ± 19	0.14 ± 0.07	−51.2 ± 1.8	-	-
I-NP	277 ± 14	0.17 ± 0.06	−39.4 ± 0.2	76.1 ± 2	77.3 ± 3
I-NP-PEG	263 ± 19	0.15 ± 0.05	−38.9 ± 2.3	81.1 ± 6	84.8 ± 3

**Table 2 pharmaceutics-14-00039-t002:** Main pharmacodynamic parameters after a subcutaneous administration of insulin (5 IU/kg) or after oral administration of bare nanoparticles loaded with insulin (I-NP; 50 IU/kg) or PEG-coated nanoparticles loaded with insulin (I-NP-PEG; 50 IU/kg). AAC corresponds to the area above the glycemic curve; T_max_ corresponds to the time when the maximum effect was achieved; C_min_ corresponds to the minimum levels of glucose (as a % of the initial values) were achieved; PA corresponds to the pharmacological activity. Data represent the mean ± SD (*n* ≥ 6). ***: *p* < 0.001, compared to I-NP.

Treatment	Dose (IU/kg)	AAC (μg/hmL)	T_max_ (h)	C_min_ (% of Initial Values)	PA (%)
Ins sc	5	1789.8 ± 155.5	3.1 ± 0.9	22.4 ± 8.7	100
I-NP	50	845.9 ± 357.6	5.1 ± 0.9	56.7 ± 15.6	4.7 ± 1.9
I-NP-PEG	50	1267.3 ± 297.0	5.6 ± 0.7	31.9 ± 20.1	14.9 ± 1.6 ***

**Table 3 pharmaceutics-14-00039-t003:** Pharmacokinetic parameters of insulin administered either subcutaneously or orally. Ins sc: subcutaneous insulin (5 IU/kg); I-NP: insulin-loaded bare nanoparticles (50 IU/kg); I-NP-PEG: insulin-loaded PEG-coated nanoparticles (50 IU/kg); C_max_: maximum plasma concentration; Tmax: time when the maximum plasma levels of insulin were achieved; AUC: area under the curve; Fr%: oral bioavailability relative to subcutaneous administrations. Data expressed as mean ± SD (*n* = 6). *: *p* < 0.05 compared to Ins sc. **: *p* < 0.01 compared to Ins sc.

Treatment	Dose (IU/kg)	C_max_ (ng/mL)	T_max_ (h)	AUC (ng/hmL)	Fr%
Ins sc	5	5.87 ± 2.07	1.37 ± 0.48	15.86 ± 4.13	100
I-NP	50	2.10 ± 0.93 **	6.00 ± 0.00	6.65 ± 2.74 *	4.2
I-NP-PEG	50	4.02 ± 1.67	5.49 ± 0.90	16.18 ± 9.76	10.2

## Data Availability

The data presented in this study are available on request from the corresponding author.

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
