# Peer review of "Zein-Based Nanoparticles as Oral Carriers for Insulin Delivery"

_pharmaceutics, 2021, doi:10.3390/pharmaceutics14010039_

Round 1

Reviewer 1 Report

The article (pharmaceutics-1504755) entitled “Zein-based nanoparticles as oral carriers for insulin delivery” by Cristian Reboredo, Carlos J. González-Navarro, Ana Luisa Martínez-López, Cristina Martínez-Ohárriz, Bruno Sarmento, and Juan M. Irache reports an application of Zein protein in oral delivery of insulin. This account is interesting for the readers of the journal. If scientific soundness is addressed, this paper can be a significant contribution to the literature. Before acceptance, there is a need for minor revision.

Specific comments:

  1. In abstract, define long forms of GRAS, PEG, C. elegans.
  2. The introduction section is sound and well presented. Please define, state the motivation behind the use of PEG.
  3. Section, 2. Materials and Methods, Consider examining the stability of insulin-loaded Zein nanoparticles.
  4. 6.1. Strain and culture conditions, define the significance of this experiment,
  5. 7.2. Induction of diabetes, examine the damage, survival of insulin-secreting cells in the pancreas.
  6. Section 3. Results; Describe stability of insulin in loaded nanoparticles against proteolytic enzymes.
  7. The results described for Pharmacokinetic and pharmacodynamic analysis are not enough, add more details.
  8. in vivo evaluation of the hypoglycemic activity of insulin-loaded nanoparticles in diabetic rats; title is descriptive, making it precise and short
  9. In discussion, address current constraints and challenges in the commercialization of such products.
  10. Present summary in separate section “Conclusions”

Author Response

Thank you very much for reviewing our manuscript. 

Reviewer 2 Report

The manuscript entitled “Zein-based nanoparticles as oral carriers for insulin delivery” co-authored by Cristian Roboredo et al. describes the preparation and characterization of insulin loaded zein nanoparticles for oral administration delivery. The methods used for preparation and detailed physicochemical and biopharmaceutical characterization of the developed insulin-loaded nanoparticles are modern and adequately selected, and the obtained results are interesting, well presented and supported by convincing discussion.

However, some details need to be clarified before the article is published:

  1. Although particles are well characterized in vitro and their effectiveness for oral insulin delivery has been demonstrated in an appropriate in vivo animal model, no data are available on particle’s stability. If the authors have data from stability studies (for example, 1 month or more), it is good to include them in the article.
  2. A convincing explanation is also needed as to why in vitro experiments on the release of insulin from both types of zein-based nanoparticles were performed at room temperature instead of the physiologically relevant oral route of 37 degrees.

Author Response

(The authors gave the same response as above.)

Round 2

Reviewer 1 Report

-